# Study on the Spatial–Temporal Pattern and Driving Mechanism of Tourism Eco-Security in the Yellow River Basin

**DOI:** 10.3390/ijerph20043562

**Published:** 2023-02-17

**Authors:** Junyuan Zhao, Shengjie Wang, Jiayue Li

**Affiliations:** 1School of Business, Xinyang College, West Section of Xinqi Avenue, Shihe District, Xinyang 464000, China; 2College of Geography and Environment Science, Northwest Normal University, Lanzhou 730070, China; 3School of Business, Central South University of Forestry and Technology, Changsha 410004, China

**Keywords:** tourism eco-security, spatial–temporal evolution mode, DPSIR model, influencing factors, the Yellow River basin

## Abstract

Tourism eco-security evaluation is an effective tool for facilitating the coordinated and sustainable economic and environmental development of tourist destinations. Based on system theory, this study established a comprehensive evaluation index system for the DPSIR model, applying the entropy–TOPSIS method, spatial autocorrelation, spatial econometric model and geo-detector to investigate the spatial and temporal evolution and drivers of tourism eco-security of the Yellow River basin. The results showed that the tourism eco-security of the Yellow River basin steadily and significantly increased from 2003 to 2020, reaching a peak in 2019, while there was a low level of overall tourism eco-security and improvement possibility. The results show a spatial evolution pattern of expansion from provincial capital cities to nearby prefecture-level cities from the middle and lower reaches to the middle and upper reaches, with significant spatial clustering and spillover effects. Factors affecting the tourism eco-security of the Yellow River basin vary in and between regional basins. Because there are many influencing factors, the key factors were further identified by spatial effect decomposition. The results of this study have important theoretical and practical value in promoting the coordinated and sustainable development of the tourism economy and ecological environment in the Yellow River basin.

## 1. Introduction

In recent decades, the sustainable development of the tourism industry has received widespread attention from various countries and regions worldwide [1,2,3]. This is primarily because the tourism industry combines the dual effects of ecological and environmental protection and green economic growth compared to traditional industries, which, to some extent, can balance the contradiction between economic development and environmental protection [4]. Although the ecological environment has suffered unprecedented threats and damage in the promotion of economic development [5], development is still the primary goal of many countries’ strategies in the context of the increasingly tense international situation and the well-being of their people. Therefore, countries must develop their tourism and other green economy industries sustainably. As the largest developing country in the world, China is experiencing a significant conflict between economic development and ecological protection [6]. In response to this problem, the tourism industry has become the main engine for governments to support China’s national ecological civilization [7,8]. However, with the rapid development of tourism and the advent of the era of mass tourism, the overburdened tourism market has highlighted the unity of opposites between the intrinsic attributes of tourism; namely, environmental dependence and resource consumption [9]. At the same time, the green development of the tourism industry is decreasing and the environmental problems closely related to tourism are becoming more serious. It has even been suggested that tourism is no longer clean [10]. In the future, global carbon emissions from tourism transport may account for 5.3% of total CO_2_ emissions [11]. If such environmental pollution damage exceeds the threshold of eco-security acceptable to a regional environment, it will threaten the region’s ecological service function and sustainable development [12]. Therefore, constructing and clarifying the relationship between tourism and the ecological environment has become an urgent practical problem for tourism scholars.

Eco-security is an important area of research on the sustainable development of tourist destinations [4,12,13]. The concept of tourism eco-security is derived from the concept of eco-security, which is an integral part of regional eco-security and an essential way to measure the sustainable development status of tourist destinations [14]. Although academia has not reached a consensus on the definition of tourism eco-security, its basic connotation is generally agreed upon. It mainly refers to the maintenance of structural stability and functional diversity of tourist destination ecosystems through the rational development of tourist resources and the effective management of tourism ecosystems within a certain spatial and temporal range. Moreover, it provides a rich material base and environmental space for tourism development and maintains the coordinated and sustainable development of the complex ecosystem of natural–social–economic systems in tourist destinations [15]. Although the term ecotourism has been coined for nearly forty years, scholars have only been studying the relationship between tourism eco-security and the environment since 2001, when tourism eco-security was first examined from the perspective of tourism carrying capacity [16]. Before that, research on tourism eco-security mainly focused on the connotations of tourism eco-security and the construction of research frameworks [17]. As the interdisciplinary approach has become increasingly sophisticated, the study of tourism eco-security has been integrated with system dynamics [18], ecology [19], geography [20], management [21], environmental science [22], and many other disciplines. Cross-disciplinary research of tourism eco-security can reveal more scientific problems and play an increasingly important role in the coordinated and sustainable development of society and economy. The research mainly covers conceptual connotation, evaluation and measurement, spatial and temporal differentiation, impact mechanism, trend prediction, spatial effect, and dynamic simulation [9]. With the continuous progress of science and technology, the research content involved will be more abundant. The scales of study include the state [20], provinces [23], cities [24], counties [25], and scenic areas [12]. Due to the varying difficulty of data acquisition, there are certain differences in the depth of research at different levels. The research objects are mainly natural landscapes such as forests, grasslands, plateaus, lakes, wetlands, watersheds, coastal areas, and islands [4]. This is related to the important ecological value of these natural landscapes. In terms of research method selection, researchers have used the CSAED (carrying capacity—support—attraction—extension—development) model, TQR (threat—quality—regulation) model, IRDS (institutional—regulatory—disturbance—security) model, PSR (pressure—state—response) model, and DPSIR (driving force—pressure—state—impact—response) model. These models and the hybrid models that improved on them were used to construct the evaluation index systems [14,26,27,28]. They used an improved TOPSIS, ecological footprint model, AHP (Analytic Hierarchy Process), composite index, and linear weighting to measure tourism eco-security quantitatively. Moreover, they identified the influencing factors of tourism eco-security through the obstacle degree model, gray correlation model, geo-detector, and spatial measurement model [9,29,30]. These diverse studies have led to a more mature system of tourism eco-security research, which provides important references for this study and other subsequent studies.

However, there are still some deficiencies in studies related to tourism eco-security and modules that should be thoroughly researched. First, established studies mainly use statistical data to select evaluation indexes, and do not apply monitoring data related to tourism eco-security [6]. Monitoring data such as vegetation coverage can better represent the real situation of a region’s ecological environment. In this study, the normalized vegetation index was added to the evaluation system to make our results more representative. Meanwhile, due to the difficulty and time cost of data acquisition, the spatial resolution of larger-scale research is low, while the spatial scope of higher-spatial-resolution research is small. Secondly, when research involves a large area, the cities included in the research vary greatly in scope. Like provincial capitals and prefectures, tourism eco-security varies greatly among regions and cities, and they should be examined separately in exploring the characteristics of spatial and temporal differentiation. Lastly, previous studies have mainly identified the influencing factors of tourism eco-security from a single approach, while this study combined the influencing factors and spatial spillover effects to identify the key influencing factors that contributed to spatial spillover effects. Therefore, the results of this study are more reliable.

In summary, the Yellow River basin, which is ecologically fragile, was selected as the object of this study, and an evaluation index system was constructed based on the DPSIR model. Furthermore, the entropy–TOPSIS method was used to evaluate the tourism eco-security index, and the spatial and temporal variation characteristics of tourism eco-security were analyzed utilizing mathematical statistics and spatial analysis. Then, the geo-detector method was used to explore the factors influencing tourism eco-security, and the key factors leading to the spatial spillover effects of tourism eco-security in the Yellow River basin were further screened by the spatial Durbin model. These findings will provide scientific theoretical references and policy inspirations for sustainable tourism development in the Yellow River basin.

## 2. Materials and Methods

### 2.1. Research Region

The Yellow River is the second-largest river in China and the fifth-longest river in the world, flowing through nine provinces (regions) (Qinghai, Sichuan, Gansu, Ningxia, Inner Mongolia, Shaanxi, Shanxi, Henan, and Shandong), and flows into the Bohai Sea in Kenli County, Shandong Province (http://www.mwr.gov.cn/szs/hl (accessed on 21 December 2022)). The Yellow River is 5464 km long and is divided into the upper, middle, and lower reaches with the nodes of Hekou Town in Inner Mongolia and Mengjin in Henan Province, with a basin area of 795,000 km^2^ (including 42,000 km^2^ of endorheic area). For the needs of the study, Henan and Shandong provinces were categorized as downstream areas, Shanxi and Shaanxi provinces as midstream areas, and the other provinces as upstream areas. The basin area used in this study was obtained from the vector boundary data provided by the Resource and Environment Science and Data Center (https://www.resdc.cn (accessed on 21 December 2022)). The prefecture-level cities (prefectures) in the basin were selected based on the regions where the main stream flows and the main part of the administrative division, including 65 prefecture-level cities (prefectures) in 9 provinces (regions) (Figure 1). The Yellow River basin is the primary place of origin of the Chinese people, and many human constructions have emerged during its long history, such as the Western Xia Tombs, the Royal Prime Minister’s Palace, and the Longmen Grottoes. At the same time, the basin ranges from the first to the third terrace in elevation from west to east. The huge altitude differences and the different climatic environments have resulted in a rich variety of natural landscapes, such as waterfalls, block mountains, rock forests, grasslands, deserts, and glaciers, which provide resources for ensuring the development of the tourism industry in the Yellow River basin. In 2020, the total GDP of the basin’s nine provinces (regions) amounted to about RMB 25,386.1 billion, accounting for 24.99% of China’s total GDP; the total tourism revenue was about RMB 2804.1 billion, accounting for 24.03% of China’s total tourism revenue. Tourism revenue plays an important role in the development of the national economy. With the ecological protection and high-quality development of the Yellow River basin being elevated to a national strategy in 2019, it has become more critical to study the eco-security of the Yellow River basin.

### 2.2. Data Collection

In this study, the spatial and temporal distribution patterns of tourism eco-security in the Yellow River basin were evaluated, and the driving mechanisms were investigated with 65 prefecture-level cities (prefectures) in the Yellow River basin as the object of study. The data in this study were derived from the China City Statistical Yearbook, China Statistical Yearbook on Environment, China Statistical Yearbook on Tourism, the Statistical Bulletin on National Economic and Social Development, the Bulletin on Environmental Status of each prefecture-level city (prefecture), and the official government websites of the Department of Environmental Protection, Department of Culture and Tourism, and Bureau of Statistics of each province (region). The missing data were supplemented by the linear interpolation method, and the foreign exchange income in total tourism revenue was transformed using the exchange rate of the corresponding year to ensure accuracy.

The normalized vegetation index data were obtained from the MODIS MOD13A1 dataset on the LAADS DAAC Data Center on the official website of NASA (https://ladsweb.nascom.nasa.gov/ (accessed on 21 December 2022)). Furthermore, the vector boundary of each prefecture-level city (prefecture) was used to extract the maximum value of each month, and then the arithmetic average of the values for 12 months was calculated to represent the vegetation cover of each year. The pictures of tourist attractions in Figure 1 were selected based on the list of “50 Best Views of the Yellow River in China” generated from the 2018 China Yellow River Tourism Conference, in order from the upper to lower reaches of the Yellow River, with consideration to both natural and human landscapes. We obtained the coordinates of boundary points of the Yellow River basin from Google Earth, the administrative zoning from the standard map service system of the Ministry of Natural Resources of the People’s Republic of China (http://bzdt.ch.mnr.gov.cn/ (accessed on 21 December 2022)), and the vector data of the Yellow River from the Resource and Environment Science and Data Center of the Chinese Academy of Sciences (http://www.resdc.cn/ (accessed on 21 December 2022)).

### 2.3. Index System Building

Tourism activities are an integrated system project that combines natural resources, economy and society, so tourism eco-security is measured by an evaluation system that covers multiple dimensions, levels, and systems [31]. Therefore, tourism eco-security should be scientifically evaluated for natural ecosystems and the interaction between human activities, economic activities, social development, and tourist destinations. The driver–pressure–state–impact–response model (DPSIR) was developed by the European Environment Agency (EEA) in 1993 by adding driver and impact indexes to the PSR framework. It systematically and comprehensively considers the relationship between humans and the environment, and can reflect the interaction between tourism activities and the ecological environment and their impacts, thus providing feedback to human activities on behalf of humans. The model has been widely used in sustainable development evaluation [32,33], eco-security evaluation [34,35], and ecological vulnerability [19,36] and water security evaluation [37,38]. In this study, the DPSIR model was used to construct the tourism eco-security index system of the Yellow River basin. In this model, each variable is in a cyclic and orderly system, and the operation mechanism is that the development of society, economy, population, and tourism act as long-term drivers (D) of the environment. If unreasonable human activities are not controlled, tourist destinations will suffer ecological damage, environmental pollution, traffic congestion, abnormal resource consumption, and the breakdown of tourist facilities. This will apply pressure (P) to many aspects of the sustainable development of the tourist destination and the residents’ normal life, and then cause a change in the ecological environment and socio-economic status (S). Changes such as the decrease in the number of days with grade 2 air quality, the decrease in the area of parkland per capita, and the increase in the number of tourist receptions will inevitably also affect the eco-security rating and attractiveness of tourist destinations, and have positive or negative effects on the economic density and industrial structure of tourist destinations (I). In order to reduce the negative impact and continue the positive effects of promoting and maintaining the benign development of the tourism industry and the ecological environment, the relevant institutions and policy makers in the tourist destinations should be motivated to respond positively (R) to the changes in the ecological state (S), such as increasing investment in environmental governance, improving the technologies for the treatment of the three wastes (waste gas; waste water; industrial residue), and raising tourists’ environmental awareness. These measures act on the social, economic, and demographic drivers (D), or directly on environmental pressures (P), states (S), and impacts (I). These subsystems, through the reconstruction and integration of their own internal element systems, create a virtuous cycle for the overall tourism eco-security system in the region (Figure 2).

In this study, based on the model built, we integrated the previous research results [30,39,40] and the actual situation of data availability, complied with the principles of completeness, dynamism, scientificity, practicality, and relevance, and analyzed the significance of indexes. Finally, 29 evaluation indexes were selected (Table 1).

### 2.4. Research Method

#### 2.4.1. Entropy–TOPSIS Method

Researchers have commonly combined the entropy method and the Technique for Order Preference by Similarity to an Ideal Solution (TOPSIS) to compare multiple indexes and objectives, which is realistic, intuitive, and reliable [41]. It can first construct the optimal and inferior solutions according to each index, then calculate the relative proximity between each index and the optimal and inferior solutions, and finally evaluate the advantages and disadvantages of each solution according to their distances from the positive and negative ideal solutions. It is considered optimal if the solution is close to the positive ideal solution and far from the negative ideal solution. If the solution is far from the positive ideal solution and close to the negative ideal solution, it is considered inferior [42]. Therefore, this study used this method to evaluate the level of tourism eco-security in each city (prefecture) of the Yellow River basin. The specific calculation process of this method is based on the evaluation of tourism eco-security in the Yangtze River Economic Belt [43]. There are no unified standards for the evaluation and grading of tourism eco-security. Based the existing research results [20,24,30], and combined with the actual situation of tourism eco-security in the Yellow River basin, we classified the levels of tourism eco-security into 7 levels: deteriorated, risky, sensitive, critically safe, generally safe, relatively safe, and very safe (Table 2).

#### 2.4.2. Spatial Autocorrelation Analysis and Spatial Econometric Model

The basic idea of spatial autocorrelation satisfies the first law of geography [44]. It is believed that spatial data are not completely independent. In two-dimensional space, spatial data related to the geographical location are spatially dependent and spatially correlated, and its purpose is to reveal the spatial interactions of certain geographical phenomena between neighboring regions [45]. Spatial autocorrelation can be divided into global spatial autocorrelation and local spatial autocorrelation. Spatial autocorrelation can reflect the distribution characteristics of regional spatial elements at the holistic and local levels, respectively. This paper adopted global spatial autocorrelation and hotspot analysis to explore the spatial characteristics of tourism eco-security in the Yellow River basin. Moran’s I is commonly used to measure the global spatial agglomeration effect [46]. The formula is as follows:(1)$$I=\frac{N}{S_{0}}\times \frac{{\text{∑}}_{i=i}^{N}{\text{∑}}_{j=1}^{N}W_{ij}\left(x_{i}-\bar{x}\right)\left(x_{j}-\bar{x}\right)}{{\text{∑}}_{i=1}^{N}{\left(x_{i}-\bar{x}\right)}^{2}}$$
where I denotes Moran’s I, N denotes the number of units studied,$\;x_{i}$ and $x_{j}$ denote the attribute values of *i* and *j* units studied, respectively, $\bar{x}$ denotes the means of $x_{i}$, $S_{0}$ denotes the sum of all elements in the spatial weight matrix, and $W_{ij}$ denotes the spatial weight matrix of units *i* and *j*. The values of Moran’s I ranged from −1 to 1. A Moran’s I > 0 indicates a positive spatial correlation, and the larger the value, the more significant the spatial correlation and agglomeration effect. A Moran’s I < 0 indicates a negative spatial correlation, and the smaller the value, the greater the spatial variation. Otherwise, Moran’s I = 0, and the space is random.

The hotspot analysis method can detect the presence of clusters of elements with spatially similar values and distinguish whether they are high-value or low-value clusters [47]. For a positive z-score that is statistically significant, the higher the z-score, the tighter the clustering of high values (hot spots). For a negative z-score that is statistically significant, the lower the z-score, the tighter the clustering of the lower values (cold spots). The formula is as follows:(2)$$G_{i}^{*}=\frac{{\text{∑}}_{j=1}^{n}w_{i,j}x_{j}-\frac{{\text{∑}}_{j=1}^{n}x_{j}}{n}\times {\text{∑}}_{j=1}^{n}w_{i,j}}{S^{\sqrt{\frac{\left[n{\text{∑}}_{j=1}^{n}w_{i,j}^{2}-{\left({\text{∑}}_{j=1}^{n}w_{i,j}\right)}^{2}\right]}{n-1}}}}$$
(3)$$S=\sqrt{\frac{{\text{∑}}_{j=1}^{n}x_{j}^{2}}{n}-{\left(\frac{{\text{∑}}_{j=1}^{n}x_{j}}{n}\right)}^{2}}$$
where $G_{i}^{*}$ denotes z-score.

The spatial Dubin model of the spatial econometric model optimizes the basic model by adding the spatial lag term of the explanatory variable based on the spatial lag model and spatial error model [48]. It is often used to reveal the spatial spillover effects of explanatory variables [6], which considers endogenous and exogenous interaction effects. Specifically, the direct effects in the model results indicate the explanatory variables’ average effect on the region’s explained variables. Indirect effects represent the average effect on non-local regions, i.e., spatial spillover effects. The total effect is the average effect on all regions. In this study, this model was used to reveal which of the main influencing factors on tourism eco-security in the Yellow River basin showed spatial spillover effects in order to screen the key factors affecting the spatial variability of tourism eco-security in the research region. The model was implemented mainly using the R language splm package [49] and Geoda 1.20 software.

#### 2.4.3. Geo-Detector

The geo-detector is a set of statistical methods that detect spatial differentiation and reveal the driving forces behind it. It is assumed that if an independent variable significantly influences a dependent variable, the independent variable and the dependent variable should have similar spatial distributions [50]. The geo-detector can detect numeric and character-based data, increasing its applicability. It consists of four modules: Factor_detector, Interaction_detector, Risk_detector, and Ecological_detector. In this study, Factor_detector was used to examine the core factors influencing the spatial distribution of the eco-security index of the Yellow River basin and to clarify the reasons for the differences in its spatial distribution. The specific formula is as follows:(4)$$q=1-\frac{{\text{∑}}_{h=1}^{L}N_{h}\sigma_{h}^{2}}{N\sigma^{2}}$$
where q  denotes the detected value of the detection factor; h and L  denote the stratification of the variable Y or factor X; $N_{h}$ and N  denote the number of cells in the stratum h and in the entire region; and $\sigma_{h}^{2}$ and $\sigma^{2}$ denote the variance of the Y values in the stratum h and in the entire region, respectively. The value of q ranges from 0 (inclusive) to 1 (inclusive), and the greater the value, the more significant the spatial heterogeneity of Y. If the independent variable X generates stratification, it can better explain Y; otherwise, it can explain Y.

## 3. Results

### 3.1. Temporal Evolutionary Characteristics of Tourism Eco-Security

This study evaluated the tourism eco-security of the Yellow River basin in China from 2003 to 2020 based on the DPSIR model, and the results showed that the tourism eco-security of the Yellow River basin showed an overall trend of steady increase and diffusion (Figure 3), reaching its peak in 2019 during the analysis period. The differences in tourism eco-security between prefecture-level cities (prefectures) in the basin also increased over time. In particular, the differences between provincial capital cities and non-provincial capital cities became extremely significant. From the upper, middle, and lower reaches of the Yellow River basin (Figure 4), all regions showed a significant upward trend, with the middle reaches showing the strongest upward trend in tourism eco-security over time (0.107/10a), followed by the lower reaches (0.082/10a) and upper reaches (0.036/10a). Compared with other regions, the level of tourism eco-security in the lower reaches is higher than that in the middle and upper reaches in 77.8% of the analysis period, showing a high level of aggregation. Tourism eco-security in the middle reaches shows a low level of dispersion in the early stage and a high level in the later stage, while the upper reaches show a low level of aggregation. On the whole, the morphology of the kernel density curves in all three regions presents a distinct right-trailing feature.

### 3.2. Spatial Pattern and Variation Characteristics of Tourism Eco-Security

This study calculated the global Moran’s I of tourism eco-security in the Yellow River basin from 2003 to 2020 using ArcGIS 10.2 (Table 3). The results showed that the Moran’s I of tourism eco-security of the Yellow River basin was positive during the study period, and the *p*-value changed from insignificant to significant (*p* < 0.05) and even reached a highly significant level (*p* < 0.01) in individual years (2017–2019). Moreover, the corresponding spatial pattern changed from random to aggregated and the trend of aggregated distribution became more pronounced over time, and the spatial correlation effect between regions was enhanced. Additionally, the aggregation effect became a fixed spatial distribution characteristic of tourism eco-security. In general, the tourism eco-security of the Yellow River basin exhibits certain spatial spillover and interaction effects, especially since 2015. We further spatially visualized the measured values of tourism eco-security during the study period based on the criteria of tourism eco-security classification (Figure 5). Due to the large span of periods and the oversized visualized sheets, 2003, 2009, 2015, and 2020 were selected as representative time nodes based on the principle of equal distribution. During the study period, the tourism eco-security levels of the Yellow River basin were between Ⅰ and V. Over time, the number of levels increased, but there were fewer prefecture-level cities (prefectures), mainly provincial capitals, within the increased levels.

On the whole, the tourism eco-security of the Yellow River basin is worrying since it is extremely insecure, but with a positive development momentum. Over time, the levels of prefecture-level cities (prefectures) around the provincial capital cities also increased, showing the spillover effect of the provincial capital cities expanding to the neighboring cities (Figure 5). In terms of the trend of the tourism eco-security index, all the prefecture-level cities (prefectures) in the Yellow River basin show an increasing trend, but the areas of rapid increase are mainly concentrated in the middle and lower reaches of the Yellow River, especially in the Shanxi, Shaanxi, and Henan provinces. In order to better reveal the local spatial characteristics of tourism eco-security in the Yellow River basin, we conducted hotspot analysis using Arcgis 10.2 software and visualized the spatial characteristics (Figure 5). The hot spots and cold spots of tourism eco-security from 2003 to 2020 have significant spatial dependence; the hot spots are mainly concentrated in the lower reaches of the Yellow River basin, and the cold spots are concentrated in the upper reaches. Over time, the range of hotspot areas expanded to the middle reaches of the Yellow River basin, the number of extremely significant areas also increased, and the number of hotspots showed a significant increasing trend (*p* < 0.01). Compared with the hotspots, the overall range of cold spots did not change much, but the number of highly significant areas increased.

### 3.3. The Driving Factors of Tourism Eco-Security

In the context of China’s green and sustainable economic development, the main drivers of tourism eco-security in the Yellow River basin need to be investigated in order to improve the level of tourism eco-security in the Yellow River basin so that tourism activities and ecological conditions can be developed with a high level of harmony. In this study, a geo-detector model was used to examine the effects of 29 evaluation indexes on the safety of the entire basin and 3 regional basin systems (Table 4). The results showed significant differences in the effects of each detection factor on tourism eco-security for the entire basin and between regional basins. In terms of the basin as a whole, the number (expressed as a percentage) and the degree of influence (expressed as a significance level) of the main influencing factors on tourism eco-security varied within the five criterion layers. The importance of the five criterion layers in affecting tourism eco-security was ranked by combining the number and degree of impact within the criterion layer and as a percentage of all indexes. The highest was Pressure (87.5%, 24.1%), followed by Impact (60%, 10.3%), State (60%, 10.3%), Driver (40%, 6.9%), and finally Response (33.3%, 6.9%). The combination of the criterion layer and the index layer shows that the seven indexes in the Pressure layer, except for population density, have significant effects on the tourism eco-security of the Yellow River basin, and the ecological conditions of tourist destinations, transportation conditions, and tourist density play an essential role in tourism eco-security. The Impact and State layer indexes affecting tourism eco-security are mainly tourism categories consisting of tourism revenue, number of tourists received, and tourism economic density. In addition, the vegetation index in the State layer greatly influences the tourism eco-security of the whole basin. The indexes of the Driver layer affecting tourism eco-security are mainly economic categories consisting of GDP per capita and disposable income per capita; in the Response layer, it is mainly environmental governance and human resource training that play a role in the tourism eco-security of the Yellow River basin. Compared with the whole basin, there are differences in factors influencing tourism eco-security between regional basins. Among them, the upper and lower reaches have an increased influence on the tourist growth rate in the Driver layer. In the Pressure layer, the number of indexes affecting tourism eco-security decreases in the upper reaches, and the indexes affected in the lower reaches are also slightly different. In the State and Response layers, the middle and lower reaches add the effects of parkland area per capita and education expenditure as a share of GDP, respectively. In the Impact layer, only the middle reaches are influenced by the share of value added of tertiary industry in GDP.

## 4. Discussion

### 4.1. Attribution Analysis of the Spatial and Temporal Variation Patterns of Tourism Eco-Security

The spatial and temporal patterns of tourism eco-security contain both intrinsic driving mechanisms and external influencing factors. The tourism eco-security of the Yellow River basin is generally at a low level, mainly due to the following reasons: First, the lower level is directly related to the classification criteria of tourism eco-security. In order to compare the results of our study with those of most studies, we chose the most frequently used classification criteria to the greatest extent possible. However, in terms of the values of the evaluation results, the results of this study are generally consistent with the evaluation results of the municipal-scale tourism eco-security of both provincial capitals and general prefecture-level cities. Tourism security is generally low, but provincial capitals are much safer than general prefecture-level cities in tourism [23]. Secondly, low eco-security is associated with the reception capacity of Chinese cities and the tourism demand and characteristics of the residents. Related studies show that China’s tourism demand has a significant positive spatial spillover effect and is still in a rising stage, and there are large differences between tourists of different ages in their travel needs along with huge differences in travel time [51]. As is known, the peak tourism periods of China’s residents are mainly concentrated in the winter and summer vacations, May Day and National Day holidays. The huge tourism impact caused by such temporal imbalance and the low tourism reception capacity (the influencing factors of the Pressure layer with the highest percentage) lead to the imbalance between tourism and the ecological environment. Despite the low tourism eco-security of the Yellow River basin, a significantly increasing trend can be seen. This result is closely connected with China’s rapid economic development and urbanization. In addition, China’s governments at all levels have introduced policies related to eco-environmental protection and residents’ awareness of “green travel” and “environmental protection” has been increasing (the share of environmental protection investment in GDP, the share of education expenditure in GDP, and the number of college and graduate students in school are the influence indexes of the Response layer). They also play an essential role in improving tourism eco-security in China.

This study shares the same results with many regional studies in China regarding variation trends. A study on the evaluation of tourism eco-security in 31 provinces of China from 1998 to 2018 revealed increasing tourism eco-security in all provinces [20]. The same results were found in the studies of tourism eco-security in the Yellow River basin in the Gansu and Hubei provinces [23,39,52]. For the middle reaches of the Yellow River, where the trend of tourism eco-security index is the strongest, ecological management primarily based on the project of returning cropland to forests and grasslands is the main reason for the variation (the normalized vegetation index has a significant effect). This region is located in the hinterland of the Loess Plateau, and its long history of development has led to sharp conflicts between people and land and the serious destruction of the ecological environment, so the level of tourism eco-security was lower than that of the upper and lower reaches at the beginning of the study period. It is noteworthy that the tourism eco-security index in the research region showed a significant decline in 2020, mainly due to the COVID-19 pandemic from December 2019 onwards. The COVID-19 pandemic forced many tourist destinations to cease operations following the lockdown measures and travel bans, especially China’s response to the outbreak, which brought tourism to a standstill [53].

Regarding spatial variation, tourism eco-security in the Yellow River basin presents a spatial distribution pattern of spillover effects from provincial capitals to nearby prefecture-level cities and a gradual transition from hot spots in the lower reaches to cold spots in the upper reaches. In the process of urbanization, provincial capitals have concentrated a large number of resources and talents, and these resources and talents will spread to the neighboring prefecture-level cities as the markets of provincial capitals become saturated, which is the radiation effect of large cities on small cities. Thus, high tourism eco-security is mostly found in and around economically developed metropolitan cities (GDP per capita and disposable income per capita are the influencing factors of the Driver layer). The same conclusion was reached in a study on tourism eco-security in the Yangtze River Delta [17]. From a perspective of the river basin, the lower reaches of the Yellow River in Eastern China are more densely populated, more economically developed and more urbanized, and have higher tourism eco-security levels than the middle and upper reaches. Just like the radiating effect of large cities, the radiating effect of the lower reaches on the middle reaches is higher than that of the upper reaches. Therefore, the hotspot areas in the middle and lower reaches are expanding continuously.

### 4.2. Key Factor Identification of the Spatial Spillover Effect of Tourism Eco-Security

Using the geo-detector model, we identified the indexes affecting the tourism eco-security of the Yellow River basin (Table 4). In terms of the number of indexes, more than half had significant effects on tourism eco-security in the research region, which indicates that the indexes we selected were highly justified. However, for the study, we assessed the tourism eco-security of the research region and identified the key influencing factors to provide a scientific basis for future planning and management [17,54]. Therefore, we used the spatial Durbin model to further screen the influencing indexes for significant spatial spillover effects, and then identified the key factors affecting the spatial evolution of tourism eco-security in the Yellow River basin (Figure 6). The results showed that the overall effect of tourism eco-security in the Yellow River basin was mainly positive, and the total effect was stronger than both the direct effect and the spillover effect, indicating that each influencing factor had a strong influence on the tourism eco-security of the Yellow River basin as a whole. The comparisons revealed that the areas with significant total and direct effects were similarly distributed, indicating that their respective regional factors mainly influence the tourism eco-security of the Yellow River basin at this stage. From the spatial spillover effects that we highlight, disposable income of the population (D2), the share of total tourism income in GDP (S4), and the number of college and graduate students (R6) are the main influencing indexes of the spillover effects. Additionally, they are the key influencing factors driving the spatial evolution pattern of tourism eco-security in the Yellow River basin (the provincial capital cities drive the neighboring prefecture-level cities). Among them, D2 has a moderately significant spillover effect across the basin due to the insignificance of the upper reaches, suggesting that the disposable income of residents in Western China is more influential than S4 and R6 in driving the spatial evolution pattern of tourism eco-security across the basin. We should primarily consider these factors when planning and managing tourism eco-security in the Yellow River basin.

From different regions of the Yellow River basin, there are large differences between regions, and the number of influencing factors tends to decrease from the lower to the upper regions of the basin, which may be related to each region’s economic development level. The spatial spillover effect is not even significant in the upper reaches, which indicates that there is still much room to improve tourism eco-security in the upper reaches. Some indexes have reached significant effects in the Driver, State, and Response levels in the middle reaches. In particular, the vegetation index had a significant effect only in the middle reaches, which indicates that the effectiveness of afforestation in this region contributes significantly more to the spatial evolution pattern of tourism eco-security than in the upper and lower reaches. It also reflects the significant positive effect of afforestation on the ecological environment and related areas. Compared with the middle and upper reaches, the most significant influence indexes were found in the lower reaches, which indicates that a better system of tourism eco-security has been formed in the lower reaches, which may be the reason for the rapid expansion of hotspots in the lower reaches compared to the middle and upper reaches. In the coming period, the tourism eco-security level in the lower reaches of the Yellow River basin is likely to expand from the provincial capital cities to the nearby prefecture-level cities much faster than in the middle and upper reaches. Therefore, when focusing on planning and managing tourism eco-security in individual areas of the Yellow River basin, both the whole basin’s and the local situation should be taken into account. Especially in the lower reaches of the Yellow River, we should regard D2, S4, and R6 as the key factors of the whole basin. Additionally, we should also consider the magnitude of the influence of other factors such as P3, P4, S2, and R1. This complicated and systemic approach makes it more difficult for planners to plan the area.

The ecological environment of the Yellow River basin is extremely fragile due to its unreasonable historical development, and the environmental carrying capacity of many regions is unable to meet the impact of tourism activities. Especially in recent years, the demand for tourism has increased rapidly. Through the measurement of the Yellow River basin tourism ecological security, this study provides a reference for the country and people of the Yellow River basin tourism ecological security temporal–spatial status quo and change trends. From the national level, although tourism ecological security in the Yellow River basin has shown a rising trend in recent decades, its overall level is still very low and not optimistic. Therefore, more emphasis should be placed on the protection of the ecological environment in the formulation of tourism development planning of the Yellow River basin, adhere to local conditions, fully consider the differences between big and small cities and different river basins, establish a cross-regional coordination mechanism for tourism ecological compensation, and effectively promote the development of tourism ecological security from the key impact factors. From the perspective of the general population, this study can highlight the important role of high-quality talent and vegetation status in improving tourism ecological security, and guide them to enhanced environmental protection awareness and to pay attention to the development of low-carbon green economy by talents. In addition, this research also has an important reference value as theoretical research.

### 4.3. Limitations and Prospects of This Study

The study of the spatial–temporal dynamic evolution pattern of tourism eco-security and its driving mechanism is to support and practice the construction of ecological civilization in China. Based on the conceptual model framework of DPSIR and the actual situation of the Yellow River basin, we selected 29 indexes to establish a comprehensive evaluation index system. We investigated the spatial and temporal evolution and driving mechanism of tourism eco-security in the region from 2003 to 2020. The study’s results may provide a scientific basis for a high level of balanced development of tourism and ecology in the Yellow River basin. However, there are some limitations of this study. First, the base unit of the study is the city, and the study’s results can only provide insights into the status and mechanisms of tourism eco-security at the municipal and provincial levels. However, we have no information on the spatial and temporal evolution of districts and counties, the smaller study units within the municipal areas. Second, although we have used the geo-detector and the spatial econometric model to identify the factors and key factors affecting tourism eco-security, it is difficult to quantify the conduction paths and mechanisms of action among these factors through conceptual models alone, and a more effective research method needs to be adopted. Related studies have indicated that the structural equation model effectively addresses such limitations [17], and we have also experimented with this method to promote the research of tourism eco-security during the research process. However, there are many problems in model construction and big data computation, so the attempt eventually failed.

However, we believe that with the continuous improvement of technical methods, it is only a matter of time before we succeed. Finally, regarding data selection, although a comprehensive index system was chosen in this study, most of the indexes are statistical, and their accuracy is greatly reduced compared with monitoring data and remote sensing observation data. This study used remote sensing observation data (normalized vegetation index). Although only one index was added, it involves a heavy workload, which may be an essential reason why such indexes are rarely used. In the future, we should increase the proportion of monitoring data or remote sensing data indexes in the whole evaluation system as much as possible, which is of great significance to increase the representativeness of the research results. In addition, in other future studies, we should try to converge the classification criteria of tourism eco-security on the same research scale as much as possible. This is not a denial of regional specificity, but rather a way to incorporate research on tourism eco-security into a system. Comparability is an indispensable prerequisite for promoting further research on tourism eco-security.

## 5. Conclusions

In this study, the DPSIR conceptual model was used to construct an index system for evaluating tourism eco-security in the Yellow River basin, and the spatial and temporal evolution patterns of tourism eco-security in the Yellow River basin from 2003 to 2020 and its driving mechanisms were analyzed. The study’s results can objectively reflect the spatial–temporal evolution process and trend of tourism eco-security. The main conclusions are as follows: First, regarding temporal changes, the overall level of tourism eco-security in the Yellow River basin is low, but it shows a steady increasing and spreading trend. There are significant differences in tourism eco-security between the provincial capitals and non-capital cities, and significant regional differences in the increasing trend and temporal state of the tourism eco-security level were seen among the upper, middle, and lower reaches of the Yellow River basin. Second, in terms of spatial variation, the tourism eco-security of the Yellow River basin has a significant and increasing spatial correlation, showing a spatial evolution pattern of expansion from the provincial capital cities to the nearby prefecture-level cities and from the middle and lower reaches to the middle and upper reaches over time, with a certain spatial spillover effect. Third, from the perspective of influencing factors, there are significant differences in the influence of each detection factor on tourism eco-security across the basin and between regional basins. The criterion layers affecting tourism eco-security from the whole basin are Pressure, Impact, State, Driver, and Response layers in descending order. The indexes with significant effects accounted for more than half of all the indexes. In order to filter out the more important indexes, we decomposed the spatial effects of these indexes with significant effects by using the spatial Durbin model. Disposable income (D2), the share of total tourism income in GDP (S4), and the number of college and graduate students in school (R6) are the main indexes of spatial spillover effects, which are also the key factors affecting the spatial evolution of tourism eco-security in the research region. In the future planning and management of tourism eco-security in the Yellow River basin at different scales, we should focus on the whole basin, fully consider the differences among different sub-basins and city levels, and effectively improve the level of tourism ecological security in a region from the key influencing factors.

## Figures and Tables

**Figure 1 ijerph-20-03562-f001:**
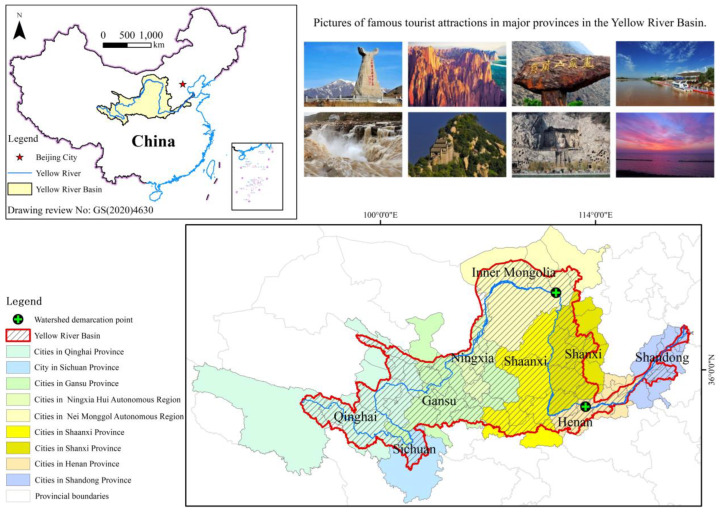
Overview of the research region in the Yellow River basin.

**Figure 2 ijerph-20-03562-f002:**
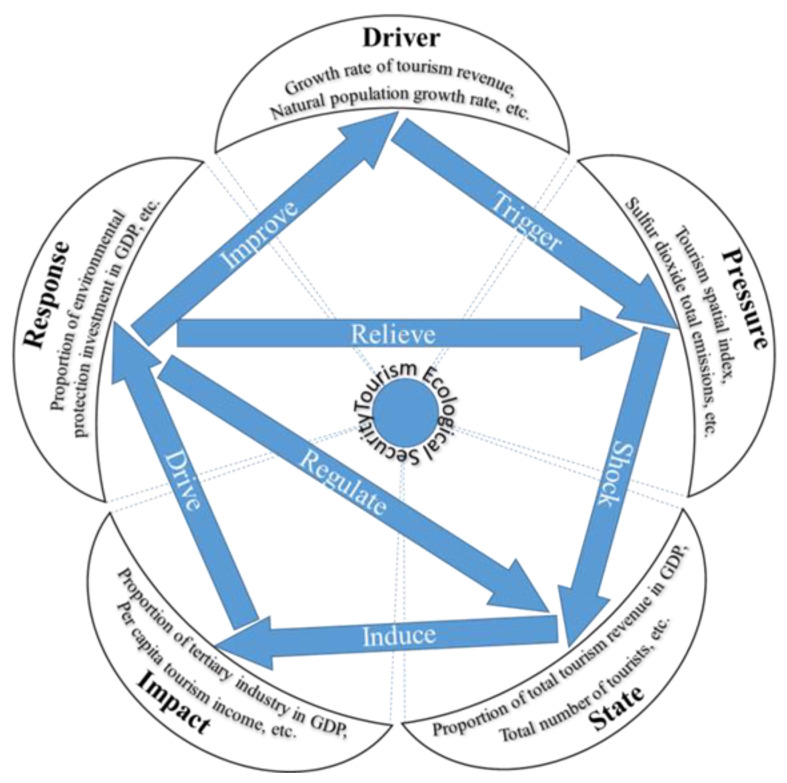
The DPSIR conceptual model for tourism eco-security evaluation.

**Figure 3 ijerph-20-03562-f003:**
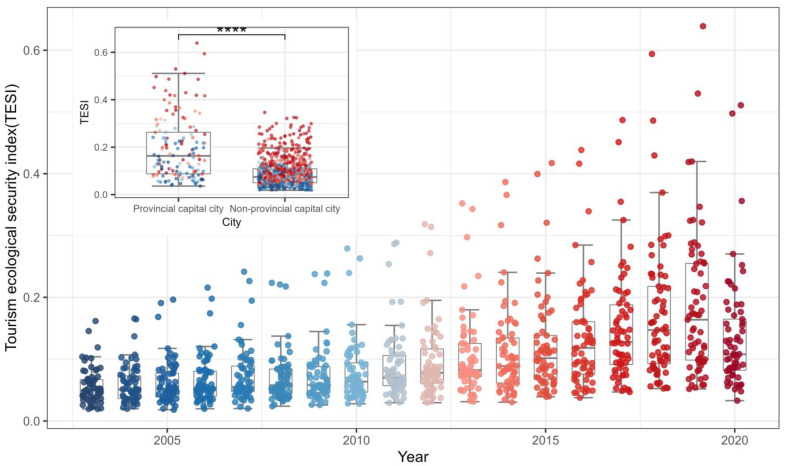
Box plot of the temporal evolution of the municipal tourism eco-security index of the Yellow River basin. **** indicates a significance of 0.0001.

**Figure 4 ijerph-20-03562-f004:**
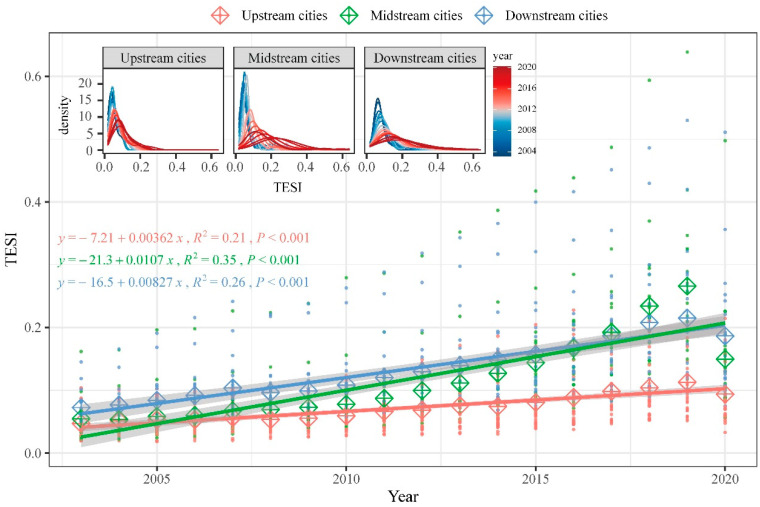
The kernel density and temporal evolution trend of the urban tourism eco-security index in the upper, middle, and lower reaches of the Yellow River basin.

**Figure 5 ijerph-20-03562-f005:**
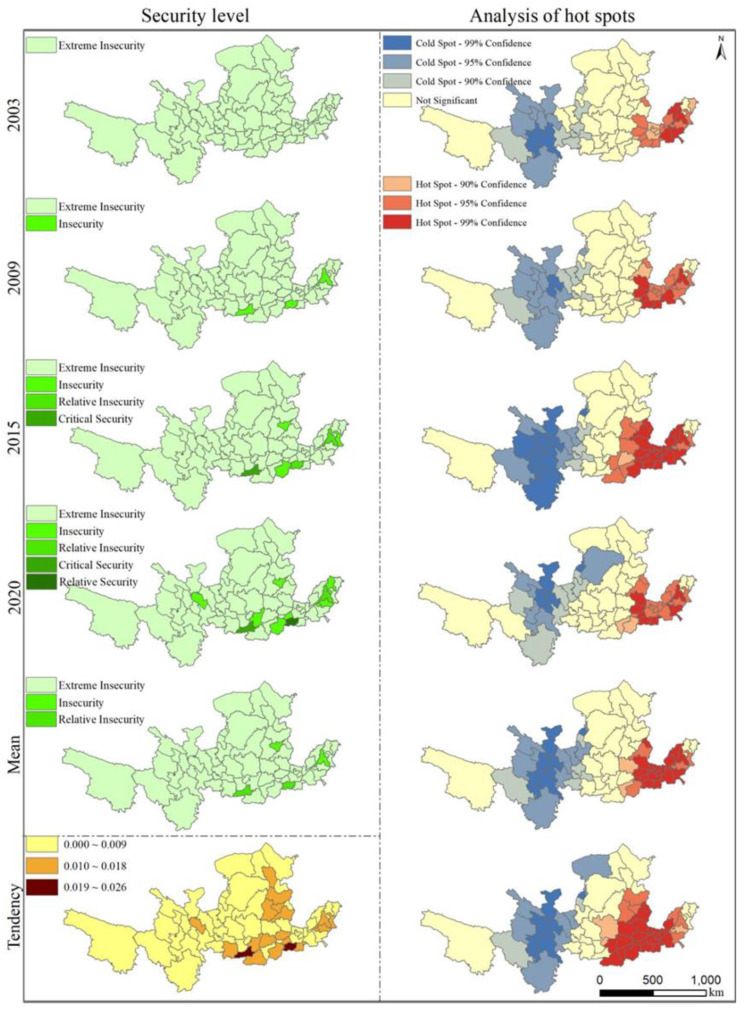
Spatial evolution and hotspot analysis of the tourism eco-security index of the Yellow River basin.

**Figure 6 ijerph-20-03562-f006:**
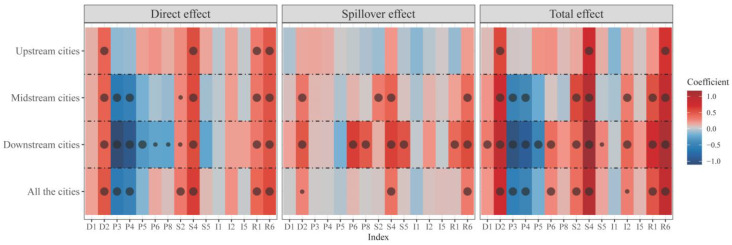
Spatial effect decomposition of the tourism eco-security index of the Yellow River basin. The large and small dots in the figure indicate the significance levels of 0.01 and 0.05, respectively.

**Table 1 ijerph-20-03562-t001:** Comprehensive evaluation index system of tourism eco-security in the Yellow River basin.

Target Layer	Criterion Layer	Factor Layer	Index Layer	Unit	Weight	The Meaning and Description of Indicators
Tourism ecological security	Driver	Economy	D1_Per capita GDP	yuan	0.026	Reflects the overall state of the regional economy.
			D2_Per capita disposal income	yuan	0.018	Reflects the living standard of local residents.
		Society	D3_Natural population growth rate	‰	0.001	Reflects the population growth trend.
		Tourism	D4_Growth rate of tourism revenue	%	0.003	Reflects the impact of tourism development and increasing tourists on the ecological environment of the tourism destination.
			D5_Growth rate of tourists	%	0.002	
	Pressure	Ecology	P1_Total wastewater discharge	10,000 tons	0.056	Reflects the potential damage caused by regional pollutant discharge.
			P2_Sulfur dioxide total emissions	10,000 tons	0.026	
			P3_Life garbage clearance volume	10,000 tons	0.047	
			P4_Solid waste production	10,000 tons	0.048	
		Traffic	P5_Tourism traffic pressure	%	0.062	Reflects the impact of the flow and influx of tourists on the transportation facilities of the tourist destination; the ratio of the number of tourists to the traffic passenger volume is used to represent it.
		Tourism	P6_Tourism spatial index	person/km^2^	0.063	Reflects the occupation of tourist space by tourists.
			P7_Population density	person/km^2^	0.033	Reflects the occupation of tourist space by local residents.
			P8_Visitor density index	%	0.038	Reflects the degree of disturbance from tourists to the life of local residents.
	State	Ecology	S1_Per capita park green area	m^2^	0.043	Reflects the quality of tourism resources.
			S2_ Normalized Vegetation Index		0.048	
			S3_The number of days with air quality above level 2	day	0.006	
		Economy	S4_Proportion of total tourism revenue in GDP	%	0.032	Reflects the development intensity of the tourism economy.
		Tourism	S5_Total number of tourists	10,000 person	0.054	Reflects the impact intensity of tourists on the tourist destination.
	Impact	Economy	I1_ Per capita tourism income	yuan	0.053	Reflects the degree of compensation from tourists to local residents.
			I2_Total tourism revenue	100_million yuan	0.060	Reflects the influence of system operation on the development level of the regional tourism economy.
		Society	I3_Proportion of tertiary industry in GDP	%	0.008	Reflects the influence of system state change on the industrial structure level of the tourism destination.
			I4_Number of employees in the tertiary industry	10,000 person	0.038	
		Tourism	I5_Tourism economic density	10,000 yuan/km^2^	0.071	Reflects the intensity of tourism development.
	Response	Economy	R1_Proportion of environmental protection investment in GDP	%	0.052	Reflects the degree of investment in the ecological security of the tourism destination.
			R2_Proportion of education expenditure in GDP	%	0.024	
		Ecology	R3_Comprehensive utilization rate of solid waste	%	0.006	Reflects the technical level of environmental protection and pollution prevention in the tourist destination.
			R4_Sewage treatment rate	%	0.005	
			R5_Life garbage treatment rate	%	0.003	
		Society	R6_The number of students in ordinary high schools	10,000 person	0.073	Reflects the quality of the population in the tourist destination.

**Table 2 ijerph-20-03562-t002:** Grading criteria for the comprehensive tourism eco-security assessment of the Yellow River basin.

Security State	Deteriorated	Risky	Sensitive	Critically Safe	Generally Safe	Relatively Safe	Very Safe
Security level	I	II	III	IV	V	VI	VII
Security index	(0, 0.2]	(0.2, 0.3]	(0.3, 0.4]	(0.4, 0.5]	(0.5, 0.6]	(0.6, 0.7]	(0.7, 1]

**Table 3 ijerph-20-03562-t003:** Global Moran’s I index of TESI.

Year	Moran’s Index	Z-Score	*p*-Value	Spatial Pattern
2003	0.0667	1.2475	0.2122	Random distribution
2004	0.0261	0.6455	0.5186	Random distribution
2005	0.0439	0.9223	0.3564	Random distribution
2006	0.0964	1.7156	0.0862	Random distribution
2007	0.1042	1.9007	0.0737	Random distribution
2008	0.0861	1.5775	0.1147	Random distribution
2009	0.0587	1.1644	0.2443	Random distribution
2010	0.0585	1.1621	0.2452	Random distribution
2011	0.0658	1.2597	0.2078	Random distribution
2012	0.1010	1.7956	0.0726	Random distribution
2013	0.0905	1.6335	0.1024	Random distribution
2014	0.1020	1.8095	0.0704	Random distribution
2015	0.1218	2.1132	0.0346	Concentration distribution
2016	0.1301	2.2195	0.0265	Concentration distribution
2017	0.1597	2.6626	0.0078	Concentration distribution
2018	0.2011	3.2872	0.0010	Concentration distribution
2019	0.2138	3.4612	0.0005	Concentration distribution
2020	0.1257	2.2055	0.0274	Concentration distribution

**Table 4 ijerph-20-03562-t004:** Analysis results of influencing factors of TESI.

Variable	All the Cities		Upstream Cities		Midstream Cities		Downstream Cities	
	qv	sig	qv	sig	qv	sig	qv	sig
D1	0.999	0.000	0.998	0.000	1.000	0.000	1.000	0.000
D2	0.997	0.000	1.000	0.000	1.000	0.000	0.997	0.000
D3	0.576	1.000	0.796	1.000	0.840	0.898	0.849	1.000
D4	0.596	1.000	0.896	0.270	0.823	0.923	0.663	0.986
D5	0.692	1.000	0.928	0.014	0.814	0.863	0.886	0.029
P1	0.918	0.004	0.735	1.000	0.896	0.170	0.990	0.053
P2	0.901	0.000	0.797	1.000	0.956	0.760	0.877	0.000
P3	0.948	0.000	0.928	0.000	0.962	0.000	0.964	0.007
P4	0.933	0.000	0.979	0.000	0.896	0.404	0.944	0.000
P5	0.999	0.000	0.994	0.000	1.000	0.000	1.000	0.000
P6	0.997	0.000	0.999	0.000	0.999	0.000	0.991	0.000
P7	0.980	0.375	0.986	0.774	0.959	0.514	0.999	0.000
P8	0.999	0.000	1.000	0.000	1.000	0.000	0.999	0.000
S1	0.978	0.243	0.816	1.000	1.000	0.000	1.000	0.000
S2	0.998	0.000	1.000	0.000	1.000	0.000	0.999	0.000
S3	0.377	0.234	0.462	0.954	0.551	0.726	0.539	0.752
S4	1.000	0.000	1.000	0.000	1.000	0.000	1.000	0.000
S5	0.992	0.000	0.987	0.006	0.996	0.000	0.996	0.000
I1	1.000	0.000	1.000	0.000	1.000	0.000	1.000	0.000
I2	0.992	0.000	0.990	0.000	0.999	0.000	0.998	0.000
I3	0.975	0.137	0.816	1.000	1.000	0.000	0.989	0.078
I4	0.762	0.925	0.699	1.000	0.889	0.200	0.593	0.997
I5	0.996	0.000	0.997	0.000	1.000	0.000	0.992	0.000
R1	1.000	0.000	1.000	0.000	1.000	0.000	1.000	0.000
R2	0.973	0.676	0.768	1.000	1.000	0.000	1.000	0.000
R3	0.421	0.967	0.645	0.260	0.616	0.798	0.383	0.906
R4	0.403	0.742	0.462	0.998	0.599	0.339	0.513	0.992
R5	0.475	1.000	0.512	1.000	0.684	0.992	0.371	1.000
R6	0.910	0.000	0.898	0.129	0.931	0.000	0.985	0.000

## Data Availability

All data used to support the findings of this research are available from the first author upon request.

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
