# Peer review of "Study on the Spatial–Temporal Pattern and Driving Mechanism of Tourism Eco-Security in the Yellow River Basin"

_ijerph, 2023, doi:10.3390/ijerph20043562_

Round 1
Reviewer 1 Report
The manuscript investigates the spatial-temporal pattern and driving mechanism of tourism eco-security in the Yellow River Basin. The topic is very interesting. The goals of this study are clearly articulated and the methods are sound. However, there are some questions that I think are important for your reference. Below are specific comments for the author.
(1) First, it is recommended to enrich the Abstract. For example, the significance of this study needs to be clarified.
(2) Second, the Introduction needs to be enriched. The Introduction section is needed to build a bit more about the research gap and rationale for this article.
(3) More information is needed to better justify the practical significance in the Discussion section.
Best of luck with revising the paper!
Reviewer 2 Report
The content of the manuscript donot in any way match the focus and scope of the journal. The said paer focuses on tourism ecosecurity which in my opinion has nothing to contribute to knowledge on public and environmental health. I strongly suspect there was an error of inclusion at some point.
I ll advise the manuscript be transferred to a more appropriate journal.
Reviewer 3 Report
i am honored to review this paper, i have following suggestions:
1. The title of driving mechanism is not suitable, it is more like driving factors, In terms of Materials and Methods, the province name could be added in figure 1. How does the tourism Eco-security is defined, which is due to ecological insecurity caused by tourism activities, local socioeconomic conditions, or local ecological envrionment carrying capacity. So, you should reflect what is the clear definition of tourism Eco-security and how to truly build a comprehensive evaluation index about tourism eco-security. Why choose to select the Yellow River Basin as a study area, what is the status quo of its tourism development, why tourism can result in such a big ecological insecurity, as your results reveal that the tourism Eco-security is below 0.7 all periods. Besides, the tourism Eco-security is more directly related with the tourism attractions, not in the city-level.
2. In terms of method, the spatial Durbin model needs to be defined. In addition, you'd better introduce the logic of the method chosen and used. Why is the spatial Durbin model and the Geo-detector used together.
3. In terms of resutls, Why are the colors in Figure 3 blue and red, but not consistent;what is the number of levels increased ?; the number of hotspots showed a significant increasing trend (P<0.01), what is the meaning of P<0.01?; As you state that TESI is divided according to "the principle of equal distribution", then extreme insecure, insecure,... types are obtained, what is the extreme insecure, insecure,...., how it can relate with the Table 2; in terms of the driving mechanism of tourism eco-security, there is just statistical description, without explaining how and why they can affect TESI. How is the importance of the five rule layers in affecting tourism eco-security ranked by combining the number and degree of impact within the rule layer and as a percentage of all indexes, for example, Response (33.3%, 6.9%), what is 33.3% and 6.9%; you mention Criterion layer ,Target layer in table 1, what is the rule layer in table 1.
4. Besides, you use the sub-system indicators to detect their impact on the composed system, there may lead to repeatitive measurement and results in inversion of causality. moreover, the geo-detector is just one year result , how it can explain the temporal evolution of TESI.
5 . In terms of discussion, it seems more like the research results, especially the figure 6, how can spatial durbin use so many influencing factors? What this means: In addition, in other future studies, we should try to converge the classification criteria of tourism eco-security on the same research scale as much as possible.
6. The conclusions shoud better be divided into several parts, besides, managerial implications should be put forward.
Round 2
Reviewer 3 Report
What practical advice can be drawn from your research?
